# Using Electrical Muscle Stimulation to Enhance Electrophysiological Performance of Agonist–Antagonist Myoneural Interface

**DOI:** 10.3390/bioengineering11090904

**Published:** 2024-09-10

**Authors:** Jianping Huang, Ping Wang, Wei Wang, Jingjing Wei, Lin Yang, Zhiyuan Liu, Guanglin Li

**Affiliations:** 1Shenzhen Institute of Advanced Technology of the Chinese Academy of Sciences, Shenzhen 518055, China; huang.jp@siat.ac.cn (J.H.); wangwei1@siat.ac.cn (W.W.); jj.wei1@siat.ac.cn (J.W.); l.yang3@siat.ac.cn (L.Y.); 2CAS Key Laboratory of Human-Machine Intelligence-Synergy Systems, Shenzhen Institute of Advanced Technology, Chinese Academy of Sciences (CAS), Shenzhen 518055, China; 3University of Chinese Academy of Sciences, Beijing 100864, China; 4Biomedical Sensing Engineering and Technology Research Center, Shandong University, Jinan 250000, China; wangpingko12345@163.com; 5The SIAT Branch, Shenzhen Institute of Artificial Intelligence and Robotics for Society, Shenzhen 518055, China; 6Shandong Zhongke Advanced Technology Co., Ltd., Jinan 250000, China

**Keywords:** agonist–antagonist myoneural interface, electrical muscle stimulation, muscle reinnervation, proprioception, electromyograph, targeted muscle reinnervation, myoelectrical prosthesis, amputation, peripheral nerve transfer, compound action potential

## Abstract

The agonist–antagonist myoneural interface (AMI), a surgical method to reinnervate physiologically-relevant proprioceptive feedback for control of limb prostheses, has demonstrated the ability to provide natural afferent sensations for limb amputees when actuating their prostheses. Following AMI surgery, one potential challenge is atrophy of the disused muscles, which would weaken the reinnervation efficacy of AMI. It is well known that electrical muscle stimulus (EMS) can reduce muscle atrophy. In this study, we conducted an animal investigation to explore whether the EMS can significantly improve the electrophysiological performance of AMI. AMI surgery was performed in 14 rats, in which the distal tendons of bilateral solei donors were connected and positioned on the surface of the left biceps femoris. Subsequently, the left tibial nerve and the common peroneus nerve were sutured onto the ends of the connected donor solei. Two stimulation electrodes were affixed onto the ends of the donor solei for EMS delivery. The AMI rats were randomly divided into two groups. One group received the EMS treatment (designated as EMS_on) regularly for eight weeks and another received no EMS (designated as EMS_off). Two physiological parameters, nerve conduction velocity (NCV) and motor unit number, were derived from the electrically evoked compound action potential (CAP) signals to assess the electrophysiological performance of AMI. Our experimental results demonstrated that the reinnervated muscles of the EMS_on group generated higher CAP signals in comparison to the EMS_off group. Both NCV and motor unit number were significantly elevated in the EMS_on group. Moreover, the EMS_on group displayed statistically higher CAP signals on the indirectly activated proprioceptive afferents than the EMS_off group. These findings suggested that EMS treatment would be promising in enhancing the electrophysiological performance and facilitating the reinnervation process of AMI.

## 1. Introduction

The prevalence of limb loss is increasing globally due to various factors, such as road accidents, diseases like diabetes, and conflicts, with the United States alone accounting for approximately 185 thousand amputees [1]. Limb loss significantly impacts an individual’s ability to perform daily tasks, affecting their overall quality of life [2]. Limb amputees have been expecting high-performance artificial upper limbs to restore the motion functions involved in their lost arms. Therefore, it is crucial to explore advanced technologies, algorithms, and surgical interventions to restore motor and sensory functions in individuals with limb loss [3,4,5,6,7].

Traditionally, body-powered mechanical prostheses were developed and used for amputees to restore their lost limb functions. With limited dexterity and indirect control mechanisms, body-powered limb prostheses are limited in utility and frustratingly slow to operate, which poses challenges in achieving intuitive functionality [8]. To improve prosthetic performance, a lot of progress has been made in the development of motorized limb prostheses that are primarily controlled by electromyographic (EMG) signals detected from the skin surface overlying the residual muscles. The most significant improvement for myoelectric prostheses should be the EMG pattern recognition based control strategy [9], in which the distinguishing characteristics of EMG patterns can be used to identify different intended movements with a trained classifier. While EMG pattern recognition based control would allow amputees to easily and intuitively operate their myoelectric prostheses, this control strategy will be inapplicable for people with above-elbow amputations because few muscles remain in their residual arm from which to extract myoelectric control signals. To address this challenge, a new neural machine interfacing technology called targeted muscle reinnervation (TMR) has been proposed and developed at the Rehabilitation Institute of Chicago, with the ability to improve control performance of multifunctional myoelectric upper-limb prostheses [10] by redirecting stump motor nerves into specific muscles to provide richer EMG signals. Previous studies have shown that the approach enabled more intuitive and parallel control of advanced multifunctional myoelectric prostheses [10,11,12]. Recently, our investigation revealed that TMR could potentially restore motor function in cases of transected median nerves, even when there was a delay in transferring them to their targets. Interestingly, we observed no significant electrophysiological disparities between immediate and delayed transfers [13]. However, it is worth noting that, despite these advancements, the subject did not experience restoration of proprioceptive feedback, such as limb position and velocity.

Recently, an innovative approach known as the agonist–antagonist myoneural interface (AMI) has emerged as a potential method to provide proprioceptive feedback. This technique leverages muscle spindle receptors located in both an agonist and antagonist muscle pair connected by a tendon [14]. In this pioneering work, Srinivasan S et al. established an AMI animal model by first implanting the tibial and peroneal nerves onto two pieces of free donor muscle, and these muscles were then connected at their ends to form an AMI model. Following a period of rehabilitation, recordings of compound action potentials (CAPs) from the nerve supplying the antagonist muscle during agonist contraction revealed the induction of proprioceptive sensations mediated by muscle spindle receptors within the artificial AMI construct. Figure 1 shows an example of the AMI concept where, when the agonist contracts, the antagonist is triggered to stretch. This method is primarily intended to simulate the situation of high-level amputees (such as shoulder disarticulation). Shoulder disarticulation patients require both targeted muscles reinnervation (TMR) and proprioceptive function reconstruction (AMI). Additionally, there is another simpler form of the AMI in which only the distal tendons of the residual muscles are connected. This simpler form of AMI does not require additional TMR and focuses solely on proprioceptive function, making it suitable mainly for lower-level amputations (such as below-knee amputations) [15,16]. Compared to the simper AMI, the more complex AMI requires TMR for both motor and proprioception function reconstruction [14,17]. In this study, only the more complex AMI is researched. As this AMI involves TMR surgery, which is fundamentally a form of peripheral nerve repair, it meant it would encounter several challenges related to peripheral nerve regeneration [14]. Firstly, in terms of TMR operation, peripheral nerves were directly transferred onto targeted muscles. This was very different from the gold standard for peripheral nerve repair in clinic (namely, neuroanastomosis) and meant the process of transferred nerve reinnervating targeted muscle in the TMR operation might have led to a failed functional recovery [18,19,20]. Moreover, those free donor muscles, cause of without neural nutrition and adequate blood circulation, would undergo atrophy [14,21]. These adverse factors collectively affect the effectiveness of proprioceptive reconstruction in AMI.

In recent decades, electrical muscle stimulation (EMS) has emerged as a dependable modality for enhancing rehabilitation following various surgeries, including nerve repair [22,23] and muscle atrophy [24,25]. These inspired us to know if the EMS treatment on the AMI muscles can significantly improve the electrophysiological performance of AMI. In this study, we aimed to investigate the potential of EMS in improving the reinnervation outcomes of AMI in animal models, along with exploring any associated electrophysiological changes.

## 2. Methods

### 2.1. Surgery

A total of 14 adult male Sprague Dawley rats, bred under Specific Pathogen-Free (SPF) conditions and weighing approximately 260 g each, were used in this study. Prior to surgery, the rats were anesthetized using 5% isoflurane in an enclosed chamber, followed by maintenance anesthesia at 2% isoflurane on an operation table equipped with a heating pad. The fur covering the bilateral hind limbs was shaved to prevent contamination during surgery. To minimize the risk of bacterial infection, the skin was sterilized using 75% medical alcohol and povidone–iodine solution. The experimental procedures were conducted in accordance with the guidelines provided by the Committee on Use and Care of Animals of the Shenzhen Institutes of Advanced Technology, Chinese Academy of Sciences (SIAT-IACUC-210907-JCS-HJP-A2050).

Figure 2A illustrates the EMS paradigm, while Figure 2B outlines the key steps involved in the AMI surgery, which include nerve implantation and tendon anastomosis. Specifically, the surgical procedure involved four incisions made on the sterilized skin. The first two incisions were made along the lateral tibia to expose and harvest the distal parts of the donor soleus muscles. The third incision was made along the left femur to expose the biceps femoris, where the two donor solei were, respectively, implanted with the ipsilateral tibial nerve (TN) and the common peroneal nerve (CPN), and connected by tendon anastomosis to construct an artificial AMI (Appendix A). The two ends of the artificial AMI were sutured onto the surface of the biceps femoris with appropriate tension. The fourth incision was made on the scalp to expose the parietal bone. At the third incision site, a connector containing two insulated stainless steel electrodes was subcutaneously pulled to the fourth incision and secured to the surface of the parietal bone using dental cement. Two stainless steel stimulating electrode ends with a 5 mm length of insulating layer peeled off were affixed onto the ends of the agonist and antagonist muscles, respectively, as depicted in Figure 2C. Meanwhile, the contralateral healthy TN and its GM, as well as the CPN and its TA, served as the natural AMI.

Following the surgical procedures, all incisions were closed and each rat received intraperitoneal injections of penicillin sodium (dose: 100,000 IU/rat) and meloxicam (dose: 2 mg/kg) for three consecutive days to prevent infection and alleviate pain. Subsequently, each rat was housed individually in a cage with a 12 h light–dark cycle in a SPF room, with free access to food and water. One day post-surgery, all rats were randomly assigned to either the EMS_on group (undergoing electrical muscle stimulation) or the EMS_off group (not subjected to electrical muscle stimulation).

### 2.2. Electrical Muscle Stimulation

Electrical muscle stimulation (EMS) was initiated following the implantation of the electrical stimulating electrodes. A tetanic contraction movement could be induced and perceived by palpation of the skin covering each artificial AMI in all rats. Beginning one day after surgery, rats in the EMS_on group received EMS treatments using NeuroTrac electrical stimulating equipment (Verity Medical, Inc., West Midlands, UK), while rats in the EMS_off group received sham treatments with the power of the stimulating equipment turned off. The parameters of the electrical muscle stimulation were set as follows: 100 Hz frequency, 200 μs pulse width, 600 tetanic contractions per day, administered for 60 min per day, six days per week, with a current intensity ranging from 2 mA to 6 mA until a proper contraction could be sensed [24]. Following the four-week EMS treatment period, an additional four-week period was allotted to allow for complete and natural recovery.

### 2.3. Neural Data Acquisition

In a healthy neuromuscular system, when the agonist muscle contracts, the relative antagonist muscle is simultaneously stretched. This stretching activates stretch-sensitive mechanoreceptors in the muscle and tendon, which then transmit impulses to the brain via ascending pathways. Consequently, an action potential related to proprioception sensation can be detected by a sensor on the peripheral nerve during this excitatory conduction process. To assess the efficacy of the constructed AMI, the study recorded the CAP of the nerve innervating the antagonist muscle during excitation of the agonist muscle by electrical stimulation. This CAP was considered a manifestation of proprioceptive sensation and may serve as a valuable additional signal for the recipient to perceive parameters such as prosthetic limb velocity and joint rotation angle.

At eight weeks post-surgery, all animals were anaesthetized with pentobarbital sodium (50 mg/kg body weight) via intraperitoneal injection. The hair covering the bilateral hind limbs was then shaved off, and the animals were placed in a prone position on a heating pad. An incision was made to expose both the artificial AMI and the natural AMI. Mineral oil was applied to prevent tissue desiccation, and the nerves supplying the muscles were carefully dissected from the surrounding tissue. During the procedure, as depicted in Figure 2D, either TNs or CPNs were electrically stimulated. Three recording electrodes were placed in contact with the agonist muscle, antagonist muscle, and the nerve of the antagonist muscle, respectively. To minimize the impact of stimulating currents on the compound action potential (CAP), the ground electrode was intentionally positioned 3 mm distally from the stimulating electrode. Additionally, to reduce the influence of muscle compound action potentials on nerve compound action potentials, the reference electrode was placed closer to the recorded nerve but farther from the muscles.

Two additional verification steps were implemented to ensure the integrity of the collected CAPs. Firstly, the tendon junction was severed to induce CAPs exclusively in the agonist muscle. Secondly, the stimulated nerve was completely crushed to verify the absence of CAPs in all sites. In this study, neural data acquisition was facilitated using the OmniPlex Express system (Plexon USA, Inc., Dallas, TX, USA) to record the evoked CAPs, while an electrical stimulator (StimuPlex Express, Plexon USA, Inc., Dallas, TX, USA) was employed to perform electrical stimulation. The stimulating current was gradually increased until a maximal action potential was observed, after which the stimulus intensity was set at 120% for recording CAPs of both the muscle and nerve. The constant parameters for evoking CAPs included a pulse width of 20 μs and a bipolar rectangle wave. Examples of recorded CAP signals from the agonist–antagonist myoneural interface in both EMS_on and EMS_off, and the healthy limb are illustrated in Figure 2E–G.

Motor unit number estimation (MUNE) is a technique used to assess the physiological status of the neuromuscular system. Arnold et al. and Gooch et al. conducted a comprehensive experimental investigation to derive a numerical estimation of MUNE [26,27]. They recorded ten grades of CAPs from the muscle and calculated the average amplitude increment to estimate the average single motor unit potential (SMUP) amplitude. The CAP of the muscle was defined as the maximal value attained when a stronger current stimulus failed to elicit a response. Subsequently, MUNE was calculated by dividing the CAP amplitude of the muscle by the SMUP amplitude.

When a transferred TN or CPN successfully reinnervates its targeted muscle, it can transmit motor impulses and evoke muscle contraction. CAPs can be recorded from the muscle when a supraliminal electrical stimulus is applied to its nerve. In this context, the propagation time refers to the duration from the onset of the electrical stimulus to the onset of the CAP, while the propagation distance is measured from the stimulus position at the nerve to the recording location at the muscle. Therefore, NCV is defined as the ratio of the propagation distance to the propagation time.

### 2.4. Statistical Analysis

To assess the statistical significance of the effect of electrical muscle stimulation (EMS) on improving the agonist–antagonist myoneural interface in the experimental group and compare it with the control group, we conducted a one-way analysis of variance (ANOVA) followed by Bonferroni post-hoc test, with a significance level *p* < 0.05. The statistical analysis was performed using the SPSS software (version 21.0, IBM Corp., Chicago, IL, USA).

## 3. Results

On average, the duration of the 14 AMI surgeries was approximately 2 h and 15 min. Furthermore, all rats successfully survived the surgery and demonstrated a gradual improvement in their physiological status throughout the rehabilitation process. Appendix A shows the targeted muscle response when electrically stimulating a nerve of the artificial AMI, low-intensity electrical stimulation corresponds to slight contraction, while high-intensity stimulation results in strong contraction.

### 3.1. Evaluation Based on Compound Action Potential (CAP)

The experimental observations depicted in Figure 3A,B reveal distinctive patterns in the CAP among healthy and artificial agonist–antagonist systems. In a healthy system, the highest CAP amplitude occurred in the agonist muscle directly evoked by electrical impulses, followed by the stretched antagonist muscle, and finally, the nerve supplying the antagonist muscle exhibited the lowest CAP amplitude. Similarly, the artificial agonist–antagonist system displayed a comparable trend, but with lower CAP amplitudes overall. However, in the artificial system, both the EMS_on and EMS_off groups exhibited lower CAP values compared to the healthy system. Notably, within the artificial system, the CAP amplitudes of both the agonist and antagonist muscles, as well as the nerve, were significantly higher in the EMS_on group compared to the EMS_off group.

Upon close examination of the results presented in Figure 3A, focusing on the stimulation of the CPN, several noteworthy observations emerge. In the EMS_on group, the CAP amplitude of the agonist muscle was significantly higher (*p* = 0.04) compared to the EMS_off group, with respective values of 1.03 ± 0.10 mV and 0.83 ± 0.05 mV. However, both groups exhibited CAP amplitudes lower than those observed in the healthy group (8.35 ± 0.40 mV). Conversely, for the antagonist muscle, the CAP amplitudes in the EMS_on and EMS_off groups were 0.23 ± 0.02 mV and 0.17 ± 0.02 mV, respectively. Notably, there was no significant difference (*p* = 0.113) between the EMS_on and EMS_off groups in terms of antagonist muscle CAP amplitude, but both groups displayed significantly lower values than the healthy group (2.84 ± 0.07 mV). Additionally, the CAP amplitude of the nerve in the EMS_on group was 0.10 ± 0.01 mV, which was significantly higher than that of the EMS_off group (0.07 ± 0.01 mV). However, both EMS groups exhibited CAP amplitudes markedly lower than those observed in the healthy group (1.58 ± 0.09 mV).

Similarly, when the TN was stimulated, as depicted in Figure 3B, notable differences emerged. In the EMS_on group, the CAP amplitude of the agonist muscle was significantly higher (*p* = 0.037) compared to the EMS_off group, with respective values of 1.09 ± 0.17 mV and 0.43 ± 0.08 mV. However, both groups exhibited CAP amplitudes lower than those observed in the healthy group (6.58 ± 0.63 mV). Turning to the results for the antagonistic muscle, the CAP amplitudes in the EMS_on and EMS_off groups were 0.55 ± 0.01 mV and 0.19 ± 0.06 mV (*p* = 0.017). Additionally, both groups displayed CAP amplitudes significantly lower than the healthy group (2.71 ± 0.80 mV). Furthermore, the CAP amplitude of the nerve in the EMS_on group was 0.21 ± 0.03 mV, significantly higher than that of the EMS_off group (0.09 ± 0.01 mV). However, both EMS groups exhibited CAP amplitudes markedly lower than those observed in the healthy group (0.80 ± 0.04 mV).

### 3.2. Evaluation Based on Motor Unit Number

Analyzing the experimental findings presented in Figure 4, it becomes evident that the highest MUNE is observed in the healthy agonist–antagonist system, while the lowest MUNE is noted in the EMS_off group. Focusing on the observations related to the CPN and its associated muscle, the EMS_on group exhibited a significantly higher MUNE value of 10.04 ± 1.34 motor units (*p* = 0.048) compared to the EMS_off group, which recorded 7.85 ± 1.21 motor units. However, both EMS groups demonstrated lower MUNE values than those observed in the healthy agonist–antagonist system (19.06 ± 0.21, *p* = 0).

Similarly, upon analyzing the results concerning the TN and its muscle, it is observed that the EMS_on group displayed a significantly higher MUNE (*p* = 0.039) than the EMS_off group. However, both EMS groups exhibited MUNE values lower than those observed in the healthy group (*p* = 0).

### 3.3. Evaluation Based on Nerve Conduction Velocity (NCV)

In the experimental data depicted in Figure 5, delineating the NCV across the EMS_off, EMS_on, and healthy groups revealed distinct trends. The EMS_off group exhibited the lowest NCV values, whereas the healthy group recorded the highest. Specifically, the average NCV of the CPN in the EMS_on group was significantly higher (30.93 ± 4.89 m/s, *p* = 0.026) compared to the EMS_off group (18.76 ± 4.14 m/s), although both were notably lower than their healthy counterparts. This trend was similarly observed in the TN.

## 4. Discussion

Limb amputation stands as the primary cause of motor and sensory impairment, and leads to significant disability. Amputation, individuals contend with the absence of motor and sensory function in the amputated limb segments originating from the excision of terminal muscles innervated by peripheral motor nerves, thus inducing motor deficits. Nonetheless, notwithstanding this loss, the central nervous system and the majority of peripheral motor nerve pathways remain functionally intact. Additionally, the severed peripheral motor nerve within the stump retains the capacity to reinnervate skeletal muscle, endowing it with renewed purpose [28,29]. To harness this potential, Kuiken et al. pioneered a TMR surgical approach, aimed at augmenting EMG signal reservoirs to enable intuitive control of myoelectric prosthetic limbs [10].

In the absence of an agonist–antagonist system post-amputation, muscle spindles in the antagonist, which are sensitive to stretch, remain inactive during agonist contraction. Consequently, amputees experience impaired proprioception, rendering them unable to recognize changes in speed, position, and posture of the prosthetic limb [30]. Clites et al. posited that AMI surgery, based on TMR principles, could restore bidirectional efferent motor signals and afferent proprioception signals concurrently. Herr’s team compared brain plasticity between 12 transtibial AMI subjects and traditional amputation (TA) subjects using functional MRI data. They found that AMI surgery induced functional network reorganization. AMI subjects exhibited significantly reduced coupling, with regions functionally dedicated to selecting attention focus in response to salient stimuli, thereby improving cognitive load management [16]. However, challenges such as muscle disuse atrophy and nerve degeneration impede AMI surgery’s ability to fully reconstruct satisfactory efferent motor and afferent somatosensory signals.

In recent decades, various treatments have emerged aimed at mitigating muscle atrophy and facilitating nerve regeneration [22,23,25]. Among these interventions, EMS has gained prominence in clinical practice, demonstrating significant advancements in addressing muscle disuse atrophy and promoting nerve regeneration. Consequently, this study explores the potential of EMS as a therapeutic modality to augment the rehabilitation outcomes of individuals with AMI.

Based on the systematic experimental protocol employed in our study, we successfully reconstructed the proprioceptive conduction pathway within the AMI. Our results revealed that the agonist muscle exhibited the highest CAP amplitudes, as it was directly stimulated, followed by the antagonist muscle, while the nerve innervating the antagonist muscle demonstrated the lowest CAP amplitude. These findings align with previous research by Clites et al. [15], which observed differential EMG amplitudes between agonist and antagonist muscles in the context of dorsiflexion and plantar flexion following AMI surgery.

Following the introduction of EMS, a discernible enhancement in nerve regeneration was observed, as evidenced by the CAP of nerves recorded in both the CPN and the TN among the EMS_on and EMS_off groups, with the EMS_on group exhibiting the highest CAP. Similarly, in terms of muscle response, a pronounced increase in CAP was noted in the agonist muscle of the EMS_on group following electrical stimulation of the TN or CPN. These findings underscore the efficacy of EMS in augmenting nerve reinnervation and facilitating proprioception restoration, thereby promoting the rehabilitation of AMI recipients. This observation aligns with the study by Fu et al., which demonstrated that adjunctive EMS application to muscles significantly improves the electrophysiological performance of denervated muscles, resulting in a stronger CAP of muscle [31].

Furthermore, similar comparative outcomes were observed in the MUNE and NCV of the CPN and TN among the EMS_on and EMS_off groups. The MUNE of the TN and CPN in the healthy group exceeded that of the TN and CPN in the EMS_on group, while the results for the TN and CPN in the EMS_on group surpassed those of the TN and CPN in the EMS_off group. Increased MUNE levels are indicative of enhanced motor function recovery, underscoring the significance of motor units in motor function rehabilitation.

This study highlights the pivotal role of EMS in the rehabilitation of AMI subjects. By applying EMS to an artificial AMI, the ischemic agonist–antagonist system experienced forced contractions, thereby decelerating its disuse atrophy process [24,26]. Additionally, Amaral et al. demonstrated that EMS could significantly enhance vessel density [32]. Elevated vessel density signifies improved physiological conditions for artificial AMI, potentially contributing to more favorable outcomes. In the past few years, Ning Ji et al. demonstrated that low-intensity FUS stimulation of the left vagus nerve effectively improved autonomic function by activating parasympathetic efferent and inhibiting sympathetic efferent, leading to reduced blood pressure. The authors believe that this novel non-invasive intervention modulation could also bring new possibilities for AMI rehabilitation [33]. In addition to the electrophysiological evaluation methods used in this study, other researchers have employed the ramp-and-hold technique to assess proprioception, identifying specializations in mechanosensory signaling and intraspinal targets for functionally identified subtypes of muscle proprioceptors in rats [34]. Furthermore, transgenic and viral tracing techniques in the neurobiology field could be applied to trace proprioceptive circuits. For example, Jacob A. Vincent used viral tracing to demonstrate that the rabies virus can infect sensory neurons in the somatosensory system and undergo anterograde trans-synaptic transfer from primary sensory neurons to spinal target neurons, delineating output connectivity with third-order neurons [35]. A more comprehensive assessment method for proprioception holds promise for a more accurate and thorough evaluation of the intervention effects of AMI.

While this study showcased the efficacy of EMS in enhancing AMI surgery rehabilitation, achieving a state of full health remains elusive. Hence, further exploration of alternative treatments is warranted to optimize rehabilitation outcomes. Additionally, although stronger proprioception signals were recorded in the peripheral nerves of EMS_on subjects, limited investigation into central nervous system changes was conducted. Therefore, future research endeavors will focus on elucidating alterations in the central nervous system and delineating the correlation between central nervous system changes and the artificial agonist–antagonist system treated with EMS.

## 5. Conclusions

This study presents the initial evidence of the beneficial impact of EMS on the electrophysiological performance of rats undergoing AMI surgery. Specifically, the CAP amplitudes of the agonist muscle and the nerve were notably higher in the EMS_on group compared to the EMS_off group. Moreover, both nerve conduction velocity and motor unit number estimation were significantly elevated in the EMS_on group relative to the EMS_off group. By inducing passive muscle contractions, EMS effectively ameliorated the physiological state of the muscles and attenuated muscle degeneration resulting from neurotrophic nutrient deficiency.

## Figures and Tables

**Figure 1 bioengineering-11-00904-f001:**
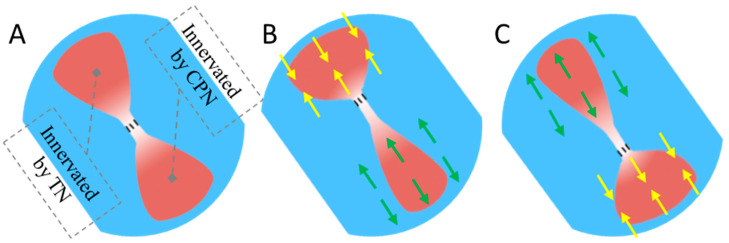
Schematics of somatosensory initiated in antagonist during agonist contraction: (**A**) an AMI implant model, (**B**) agonist innervated by tibial nerve (TN) shortens and antagonist innervated by common peroneal nerve (CPN) stretches, and (**C**) agonist innervated by CPN stretches shortens and antagonist innervated by TN. Yellow arrows indicate the direction of agonist contraction and green arrows indicate the direction of antagonist.

**Figure 2 bioengineering-11-00904-f002:**
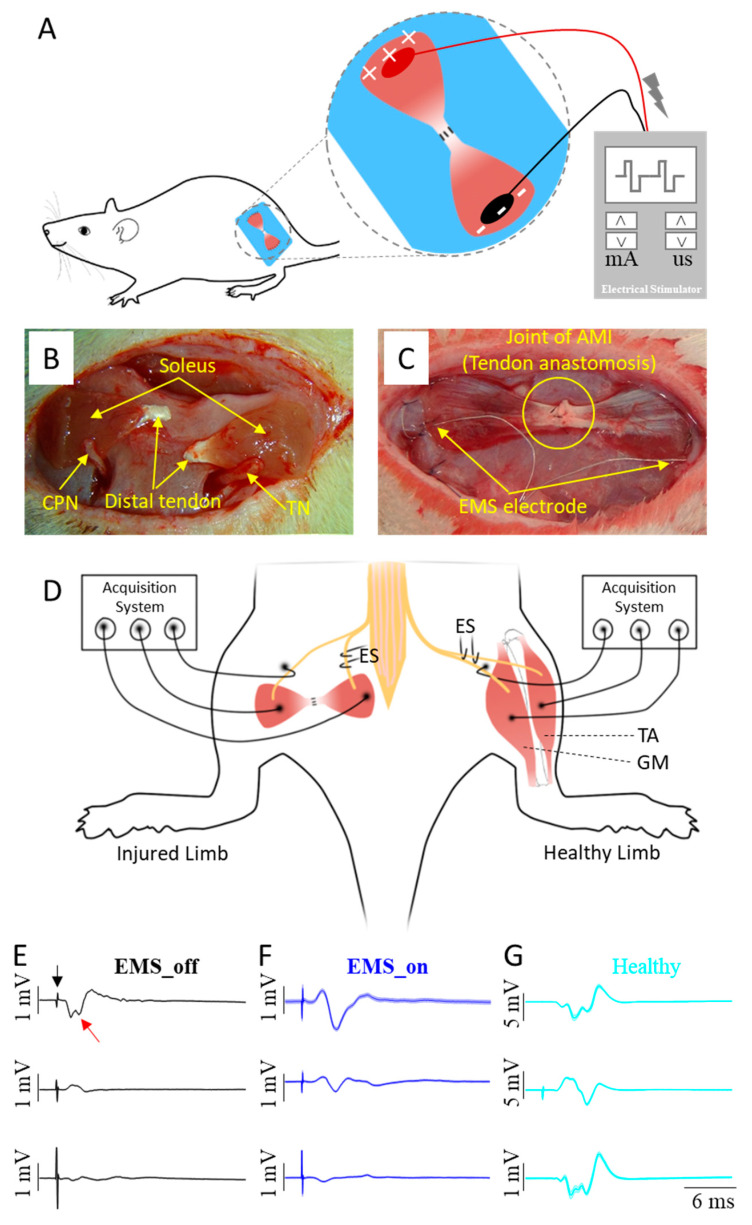
Setup of study. (**A**) Schematic of electrical muscle stimulus (EMS); (**B**) common peroneal nerve (CPN) and tibial nerve TN were, respectively, transferred onto donor soleus; (**C**) two soleus were connected by distal tendons and fixed onto biceps femoris with proper tension, and two electrical stimulating (ES) electrodes were fixed onto ends of AMI for chronic muscular EMS treatment; (**D**) recorded compound action potential (CAP) from artificial AMI and control AMI evoked by electrically stimulating nerve; and (**E**–**G**) represented CAP signals evoked by ES on CPN from agonist–antagonist myoneural interface in EMS_off and EMS_on limbs, and healthy limb, respectively. Top traces are muscle CAP signals recorded from agonist, middle ones are also muscle CAP signals recorded from antagonist, and the bottom ones are nerve CAP signals recorded from nerve innervating antagonist. Arrow in black indicates the artificial of ES, and arrow in red indicates the CAP amplitude.

**Figure 3 bioengineering-11-00904-f003:**
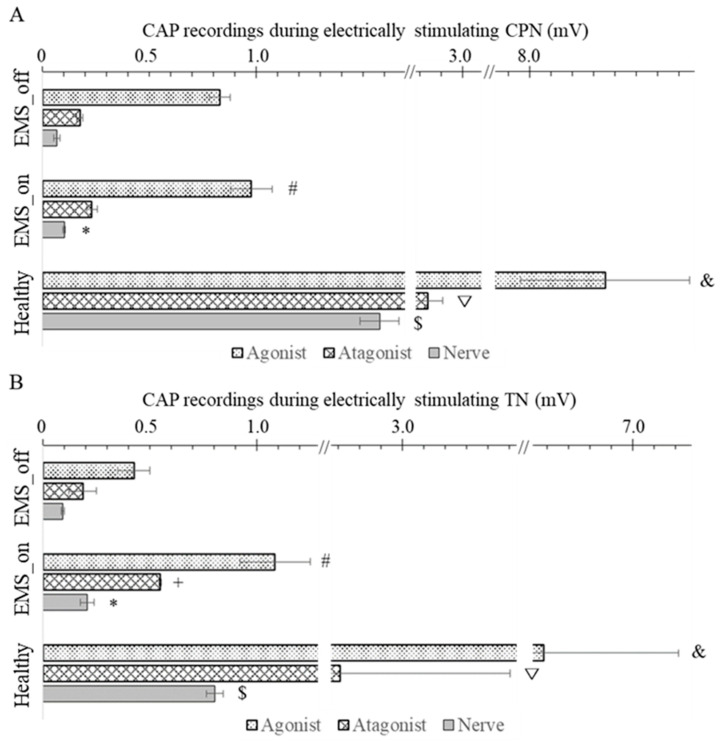
Compound action potential recordings from nerve and muscle of agonist–antagonist group. (**A**,**B**) compound action potential (CAP) during electrically stimulating common peroneal nerve (CPN) and tibial nerve (TN), respectively. *—There is a significant difference between nerve CAP of EMS_on and EMS_off; +—There is a significant difference between muscle CAP of antagonist in EMS_on and EMS_off; #—There is a significant difference between agonist CAP of EMS_on and EMS_off; $—There are significant differences between nerve CAP of healthy system and artificial agonist–antagonist system; ▽—There are significant differences between muscle CAP of agonist in healthy system and that of in artificial agonist–antagonist system; &—There are significant differences between muscle CAP of antagonist in healthy system and that of in artificial agonist–antagonist system.

**Figure 4 bioengineering-11-00904-f004:**
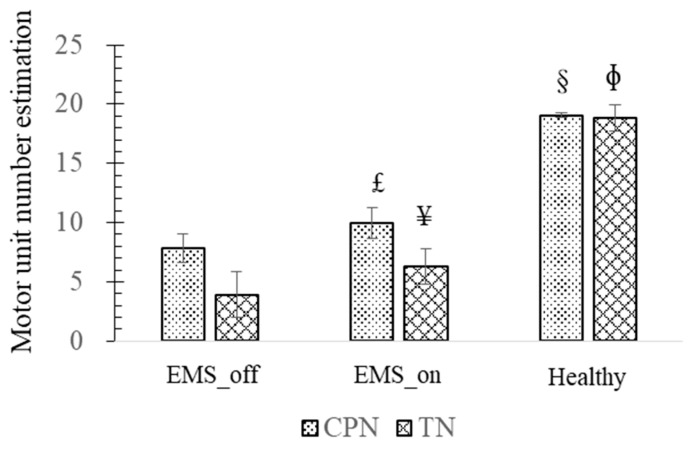
Motor unit number estimation (MUNE). £—There is a significant difference between MUNE of common peroneal nerve (CPN) in EMS_on and that of in EMS_off; ¥—There is a significant difference between MUNE of tibial nerve (TN) in EMS_on and that of in EMS_off; §—There are significant differences between MUNE of CPN in healthy and that of in artificial agonist–antagonist system; Φ—There are significant differences between MUNE of TN in healthy and that of in artificial agonist–antagonist system.

**Figure 5 bioengineering-11-00904-f005:**
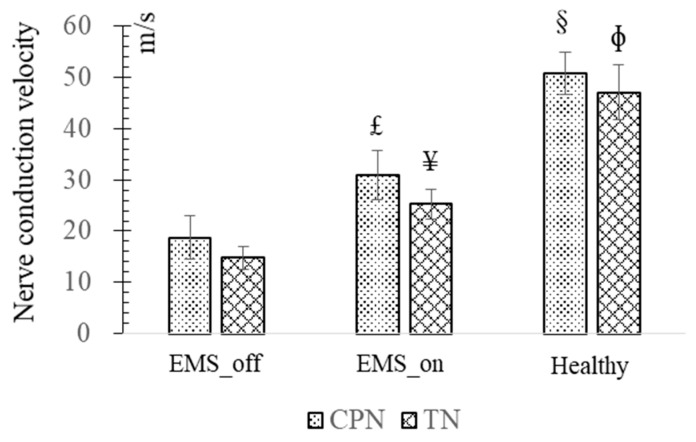
Nerve conduction velocity (NCV). £—There is a significant difference between NCV of common peroneal nerve (CPN) in EMS_on and that of in EMS_off; ¥—There is a significant difference between NCV of tibial nerve (TN) in EMS_on and that of in EMS_off; §—There are significant differences between NCV of CPN in healthy and that of in artificial agonist–antagonist system; Φ—There are significant differences between NCV of TN in healthy and that of in artificial agonist–antagonist system.

## Data Availability

The data presented in this study are available upon request from the corresponding author.

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
