# Peer review of "Using Electrical Muscle Stimulation to Enhance Electrophysiological Performance of Agonist–Antagonist Myoneural Interface"

_bioengineering, 2024, doi:10.3390/bioengineering11090904_

Round 1

Reviewer 1 Report

Comments and Suggestions for Authors

The work introduces a novel approach to limb prosthetics, significantly enhancing prosthetic movement and control through the electrophysiological performance of Myoneural Interface signals. As a practical implementation with tangible benefits for practitioners and readers of the Journal, I highly recommend this article for publication.

Author Response

Comment 1: The work introduces a novel approach to limb prosthetics, significantly enhancing prosthetic movement and control through the electrophysiological performance of Myoneural Interface signals. As a practical implementation with tangible benefits for practitioners and readers of the Journal, I highly recommend this article for publication.

Response 1: Thank you for your recognition and suggestion.

Reviewer 2 Report

Comments and Suggestions for Authors

Dear Editor and Authors,

The paper describes how Electrical Muscle Stimulation to Enhance Electrophysiological Performance of Agonist-Antagonist Myoneural Interface using rat models. Even though the experimental procedures and results were well discussed. However, literature comparison is weak, so the superiority of the approach should be discussed under the light of literature.

Kind Regards,

Author Response

Comment 1: The paper describes how Electrical Muscle Stimulation to Enhance Electrophysiological Performance of Agonist-Antagonist Myoneural Interface using rat models. Even though the experimental procedures and results were well discussed. However, literature comparison is weak, so the superiority of the approach should be discussed under the light of literature.

Response 1: Thank you for pointing this out. To address the gaps in the literature discussion, the authors have added the following three aspects: AMI plasticity, treatment, and assessment. Firstly, Herr's team compared brain plasticity between transtibial AMI subjects and traditional amputation (TA) subjects using functional MRI data. They found that the AMI surgery induced functional network reorganization. AMI subjects exhibited significantly reduced coupling with regions functionally dedicated to selecting attention focus in response to salient stimuli, thereby improving cognitive load management. Secondly, Ji N et al. demonstrated that low-intensity focused ultrasound stimulation of vagus nerve effectively improved autonomic function by activating parasympathetic efferent and inhibiting sympathetic efferent, leading to reduced blood pressure. This novel non-invasive intervention modulation may also bring value for AMI rehabilitation. Lastly, in addition to the electrophysiological evaluation methods used in this study, other researchers have employed the ramp-and-hold technique to assess proprioception, identifying specializations in mechanosensory signaling and intraspinal targets for functionally identified subtypes of muscle proprioceptors in rats. Furthermore, transgenic and viral tracing techniques in the neurobiology field could be applied to trace proprioceptive circuits. For example, Zampieri N et al. used viral tracing to demonstrate that rabies virus can infect sensory neurons in the somatosensory system and undergo anterograde trans-synaptic transfer from primary sensory neurons to spinal target neurons, delineating output connectivity with third-order neurons. A more comprehensive assessment method for proprioception holds promise for a more accurate and thorough evaluation of the intervention effects of AMI.

Reviewer 3 Report

Comments and Suggestions for Authors

This paper presented an investigation about using electrical muscle stimulation (EMS) to facilitate the electrophysiological performance of agonist-antagonist myoneural interface (AMI). The experiment was done in rats. It was found that the EMS intervention was able to elicit higher compound action potential (CAP) during stimulation of the common peroneus nerve (CPN) and tibial nerve (TN). There were also higher number of motor units and higher nerve conduction velocities in the EMS group. The results were interpreted as the evidence that EMS was effective in enhancing nerve regeneration and facilitating proprioception restoration. Overall, the paper is well-organized, and the results are clear. However, I have a few questions about how it is related to AMI. It was unclear whether the results is specific to the surgical procedure in this study, which might be quite different from the standard AMI method. Clarifications are needed for this reviewer to fairly evaluate this paper.

Introduction:

The authors mentioned several challenges for AMI, but the citations (18,19,20) are not directly related to AMI. These citations present some general findings about peripheral nerve injuries. Please make more explicit connections between the cited work and AMI, and show evidence that directly supports the argument about atrophy and reinnervation in AMI.

Method:

In the work of Clites (15), my understanding is that the distal ends of tibial and peroneus nerves were transected and RPNIs were established. They were not directly used in the AMI, which consisted of the tibialis anterior and lateral gastrocnemius muscles whose tendons were connected. In the present work, however, the CPN and TN were relocated to Soleus that function as the AMI. This surgical procedure is more similar to TMR than the original form of AMI. Please clarify the rationale of this surgical procedure in the present study, and how it compares other AMI work.

Comments on the Quality of English Language

No comments

Author Response

Comment 1: This paper presented an investigation about using electrical muscle stimulation (EMS) to facilitate the electrophysiological performance of agonist-antagonist myoneural interface (AMI). The experiment was done in rats. It was found that the EMS intervention was able to elicit higher compound action potential (CAP) during stimulation of the common peroneus nerve (CPN) and tibial nerve (TN). There were also higher number of motor units and higher nerve conduction velocities in the EMS group. The results were interpreted as the evidence that EMS was effective in enhancing nerve regeneration and facilitating proprioception restoration. Overall, the paper is well-organized, and the results are clear. However, I have a few questions about how it is related to AMI. It was unclear whether the results is specific to the surgical procedure in this study, which might be quite different from the standard AMI method. Clarifications are needed for this reviewer to fairly evaluate this paper.

Response 1: Thank you for pointing this out, and we agree with this comment. In reference to the study mentioned as reference 16, the creators of AMI established an AMI animal model by first implanting the tibial and peroneal nerves into two pieces of free donor muscle. These muscles were then connected at their ends to form AMI. This method is primarily intended to simulate the situation of high-level amputees (such as shoulder disarticulation). Shoulder disarticulation patients require both targeted muscles reinnervation (TMR) and proprioceptive function reconstruction (AMI). This study used this surgical approach to create the AMI animal model. Additionally, reference 15 describes a simpler form of the AMI, in which only the distal tendons of the residual muscles are connected. This simpler form of AMI does not require additional TMR and focuses solely on proprioceptive function, making it suitable mainly for lower-level amputations (such as below-knee amputations).

Comment 2: Introduction, the authors mentioned several challenges for AMI, but the citations (18,19,20) are not directly related to AMI. These citations present some general findings about peripheral nerve injuries. Please make more explicit connections between the cited work and AMI, and show evidence that directly supports the argument about atrophy and reinnervation in AMI.

Response 2: Thank you for pointing this out, and we agree with this comment. In reference 16, the method used by the creators of AMI to construct the AMI animal model involved first implanting the tibial and peroneal nerves into two pieces of free donor muscle. These muscles were then connected at their ends to form AMI, and finally, the AMI was fixed to the biceps femoris. The authors have validated the feasibility of proprioceptive reconstruction using AMI and have also reported muscle atrophy in the results. Secondly, in cases where both motor function and proprioceptive function reconstruction are needed, TMR is required to transfer the nerve to the targeted muscle to restore motor function, while connecting the ends of the target muscle is necessary to restore proprioceptive function. This study simulated such a scenario and, to correspond with the method described in reference 16, similarly referred to it as the AMI model. Since this model involves TMR surgery, which is fundamentally a form of nerve repair, relevant literature on nerve repair has been cited to support this approach.

Comment 3: Method, in the work of Clites (15), my understanding is that the distal ends of tibial and peroneus nerves were transected and RPNIs were established. They were not directly used in the AMI, which consisted of the tibialis anterior and lateral gastrocnemius muscles whose tendons were connected. In the present work, however, the CPN and TN were relocated to Soleus that function as the AMI. This surgical procedure is more similar to TMR than the original form of AMI. Please clarify the rationale of this surgical procedure in the present study, and how it compares other AMI work.

Response 3: Thank you for pointing this out, and we agree with this comment. Thank you for your valuable suggestions. In reference 15, your understanding is correct, as the subjects in that article were lower-level below-knee amputees. Since the residual muscles in below-knee amputees are sufficient to support intuitive control of the ankle joint, additional motor function reconstruction is not required. Only tendon anastomosis of the distal residual muscles is needed to restore proprioceptive function, with the use of RPNI to prevent neuroma formation. However, the surgical approach used in this study is similar to that described in reference 16, primarily simulating the scenario of high-level amputees (such as shoulder disarticulation). High-level amputees require both TMR for motor function reconstruction and AMI for proprioceptive function reconstruction.

Round 2

Reviewer 3 Report

Comments and Suggestions for Authors

While the authors answered my question in the response letter, no change was made in the manuscript to improve clarity.

Reference [16] should be explicitly singled out as the foundation of the surgical procedure used in this study. In the paragraph of line 78-95, the authors used [14] and [15] as the reference for the surgical model of AMI. Instead [16] should be used. Importantly, the authors should also use their response to my questions in the manuscript to discuss two types of AMI, one that does not require reinnervation and one that does (as well as their clinical use scenarios). Note that this study only targets the type of AMI that does require reinnervation. 

The author should also make clear in the paper that there is no actual clinical evidence for atrophy and reinnervation yet. The problem they are addressing is only a potential issue (an educated guess) based on common knowledge about general nerve injury.

Comments on the Quality of English Language

Generally ok.

Author Response

Comment 1: Reference [16] should be explicitly singled out as the foundation of the surgical procedure used in this study. In the paragraph of line 78-95, the authors used [14] and [15] as the reference for the surgical model of AMI. Instead [16] should be used. Importantly, the authors should also use their response to my questions in the manuscript to discuss two types of AMI, one that does not require reinnervation and one that does (as well as their clinical use scenarios). Note that this study only targets the type of AMI that does require reinnervation.

Response 1: Thank you for pointing this out, and we agree with this comment. In this reference 16 (it was reference 14 in the revised version), the proposer of AMI achieved proprioceptive function reconstruction by implanting the tibial and common peroneal nerves into two separate donor muscles. This surgical paradigm is what our study is based on. The AMI procedure varies depending on the level of amputation (reference 14 is for high-level and reference 15 is for low-level). In our study, the AMI paradigm simulates a high-level amputation scenario, it requires not only the reconstruction of proprioceptive function through AMI but also TMR to restore motor function (reference 14). Accordingly, the relevant content has been added to the new manuscript (line 88-99).

Comment 2: The author should also make clear in the paper that there is no actual clinical evidence for atrophy and reinnervation yet. The problem they are addressing is only a potential issue (an educated guess) based on common knowledge about general nerve injury."

Response 2: Thank you for pointing this out. There are few clinical studies on muscle atrophy and nerve regeneration failure after AMI or TMR surgery. Shaughnessy et al. reported a case in which a patient who underwent humeral amputation and TMR surgery failed to generate sufficiently strong EMG signals in the targeted muscles seven months post-surgery, thus failing to achieve intuitive control of the myoelectric prosthesis (reference 18 in the revised version). Srinivasan et al. reported cases of targeted muscle atrophy after AMI (reference 14 in the new revised manuscript). I have inserted these references into the new version of the manuscript and provided an explanation.